# Continuous quality improvement with a two-step strategy effective for mass SARS-CoV-2 screening at the Tokyo 2020 Olympic and Paralympic Games

Hayato Miyachi[1,2]*, Satomi Asai[1], Rika Kuroki[3], Kazuya Omi[3,4,5], Chiaki Ikenoue[6], Satoshi Shimada[7,8]

1 Department of Laboratory Medicine, Tokai University School of Medicine, Isehara, Japan, 2 Faculty of Clinical Laboratory Sciences, Nitobe Bunka College, Tokyo, Japan, 3 SRL Inc., Tokyo, Japan, 4 H.U. Group Research Institute G.K., Tokyo, Japan, 5 Fujirebio Inc., Tokyo, Japan, 6 Center for Field Epidemic Intelligence, Research and Professional Development, National Institute of Infectious Diseases, Tokyo, Japan, 7 Department of Eco-Epidemiology, Institute of Tropical Medicine (NEKKEN), Nagasaki University, Nagasaki, Japan, 8 Center for Emergency Preparedness and Response, National Institute of Infectious Diseases, Tokyo, Japan

* miyachi@tokai.ac.jp

**Data Availability Statement:** Due to a confidentiality agreement with the Tokyo Organizing Committee of the Olympic and

## Abstract

The Tokyo 2020 Olympic and Paralympic Games (Games) were held during the height of the coronavirus disease 2019 (COVID-19) pandemic. To detect people infected with severe acute respiratory syndrome coronavirus-2 (SARS-CoV-2) early enough to contain the spread and to facilitate the timely arrival of athletes at their game venues, all participating athletes staying in the Olympic Village (up to 14,000) were screened daily for the infection. Toward this aim, a two-step strategy was adopted comprising screening of self-collected saliva samples using a chemiluminescence enzyme immunoassay, followed by confirmatory testing using polymerase chain reaction. The testing system was integrated with an information management system covering all steps. To ensure the accuracy of the test results, rigorous quality assurance measures and monitoring of performance/specimen quality were implemented. A chronological chart analysis was implemented to monitor the holistic process and to give feedback to improve the sampling. Nearly all test results for 418,506 saliva samples were reported within 12 hours of sample collection, achieving the target mean turnaround time of 150 minutes for confirmatory testing. As a result, athlete activity and performance for the Games were ensured. The chronological chart confirmed that no athletes were withdrawn due to a false-positive result, and no infection clusters were identified among the athletes in the Olympic Village. In conclusion, continuous quality improvement as part of the two-step strategy for mass screening for COVID-19 contributed to the success of the Games during the pandemic. The quality practice, systems, and workflows described here may offer a model for future mass-gathering sporting events during similar major infectious disease epidemics.

Paralympic Games (TOCOG), there are strict restrictions on sharing data. Information on clinical performance can only be shared as specified in TOCOG's press releases. Sharing deidentified data would be considered inappropriate, as it could potentially reveal identifying or sensitive information about participants when combined with details from TOCOG's press releases. Releasing such data could compromise the accountability of the Committee, which played a critical role in controlling the spread of SARS-CoV-2 during the Games. As a result, the Ethics Committee of Nitobe Bunka College has imposed strict restrictions on sharing deidentified data sets. When a researcher is interested in accessing the minimal dataset for this study, please provide a rationale for the request along with details regarding the de-identified data. Data are available upon request from the Ethics Committee of Nitobe Bunka College, via mail (3-43-16 Nakano, Nakano-ku, Tokyo 164-0001, Japan) or telephone (+81-3-3381-0121), for researchers who meet the criteria for access to confidential data.

**Funding:** The author(s) received no specific funding for this work.

**Competing interests:** I have read the journal's policy and the authors of this manuscript have the following competing interests: For non-financial conflicts of interest, H.M., S.A., S.S., and C.I. have published several papers about COVID-19 and could have conflicts of interest among healthcare professionals. For financial conflicts of interest, H. M. was a consultant at SRL Laboratories and an adviser for the laboratory in the Olympic and Paralympic Village in Harumi, Tokyo, Japan. H.M. and S.A. were involved in the nationwide external quality assessment of SARS-CoV-2 nucleic acid tests, a project commissioned by the Ministry of Health, Labour and Welfare of Japan. S.A. contributed to studies on SARS-CoV-2 PCR tests developed by DENSO Corp. and KYORIN Pharm. Co., Ltd. that were financially supported by the Japan Agency for Medical Research and Development. R.K. and K.O. rendered laboratory service for athletes and team officials during the Tokyo 2020 Games as a commission project by the Tokyo Organizing Committee of the Olympic and Paralympic Games. S.S. and C. I. worked as staff members for Tokyo 2020 during the Tokyo 2020 Games. The authors report no other potential conflicts of interest.

## Introduction

Held in Summer 2021 following a 1-year postponement during the coronavirus disease 2019 (COVID-19) pandemic, the Tokyo 2020 Olympic and Paralympic Games faced the challenge of surging COVID-19 rates, with the incidence reaching unprecedented levels both in Tokyo and globally [1]. The Tokyo 2020 Infectious Diseases Control Center organized by the Tokyo Organizing Committee of the Olympic and Paralympic Games (TOCOG) played a vital role in containing the spread of severe acute respiratory syndrome coronavirus 2 (SARS-CoV-2) during the Games by implementing planned biosafety protocols, including frequent testing for SARS-CoV-2 [2].

Although the "gold standard" of SARS-CoV-2 testing by means of polymerase chain reaction (PCR) using nasopharyngeal (NP) swab samples is accurate and reliable, the turnaround time (TAT) for reporting the results may take 24–48 h. Such delays may lead to not only further transmission of disease but can also diminish the daily activity of athletes for training and arrival at the event venues. Furthermore, it requires trained professionals in full protective equipment to collect specimens individually, one person at a time. Solutions to improve the efficiency of mass screening for SARS-CoV-2 included implementing a two-step screening process comprising screening of self-collected saliva samples using a chemiluminescence enzyme immunoassay (CLEIA), followed by confirmatory PCR testing [3–5]. The scientific framework of this strategy has been reported previously. This strategic allocation of PCR tests to cases that were CLEIA-positive or inconclusive not only maintained accuracy but also significantly improved logistical efficiency in administering mass screening. To detect the virus with sufficient time to contain the spread as well as to maintain their daily training activities prior to competing in events and facilitate the timely arrival of athletes at their game venues, all participating athletes and others staying in the Olympic Village were screened daily for SARS-CoV-2 infection with this two-step strategy. Although this strategy had been confirmed to work in a short period for screening travelers at international airports, its utility for mass-gathering sporting events had not previously been evaluated. We implemented this strategy in an international multi-sports event, i.e. the Tokyo 2020 Olympic and Paralympic Games, implementing daily screening and longitudinal follow-up of all athletes.

The integration of a testing system with an information management system covering all steps from specimen collection to reporting of the results, enabled confirmatory test results to be provided within 12 hours, aiming for a mean TAT of 150 minutes for confirmatory tests. In addition, our challenge was to avoid the risk of infection when gathering at the collection booth; therefore, we adopted a policy of self-collection of saliva in individuals' own rooms in their living quarters. However, self-sampling in individual rooms posed a risk of poor specimen quality due to suboptimal preparation, collection, and storage. To ensure the accuracy of the COVID-19 screening test, we implemented continuous quality improvement through rigorous quality assurance measures, performance/specimen quality monitoring by an internal quality assessment, and a chronological chart analysis for individual cases.

## Material and methods

### A two-step mass screening strategy

The study was planned and conducted at the request of the TOCOG according to the regulation and the criteria specified by the Ministry of Health, Labour, and Welfare of Japan. Up to 14,000 athletes and team officials were tested for SARS-CoV-2 daily to detect infections as soon as possible, as required by the TOCOG.

To facilitate the timely arrival of athletes at their game venues, test results had to be available within a 12-hour TAT. For this mass screening, a two-step strategy was adopted, instead of screening by PCR alone [2–5]. First, quantitative antigen testing was performed using a quantitative CLEIA with a fully automated instrument (Lumipulse® Presto SARS-CoV-2 Ag, Fujirebio, Tokyo, Japan). The quantitative CLEIA is a high-throughput assay with high sensitivity, specificity, and short TAT, and can provide rapid screening test results using saliva specimens. All specimens were tested by CLEIA with positive and negative thresholds of 4.0 pg/mL and 0.67 pg/mL, respectively, using the specified reagents according to the manufacturer's instructions. Positive or inconclusive results were confirmed using a standard PCR test. If the saliva PCR showed positive results, a nasopharyngeal (NP) swab was collected at the fever clinic in the Olympic Village for confirmatory PCR testing using the SARS-CoV-2 Direct Detection RT-qPCR Kit (Takara Bio Inc., Kusatsu, Japan) and instruments such as QuantStudio5 (Thermo Fisher Scientific K.K., Waltham, MA, USA), LightCycler 96 (Roche Molecular Systems, Inc., Pleasanton, CA), or CFX96 (Bio-Rad Laboratories, Inc., Hercules, CA, USA). Symptomatic participants also underwent PCR testing of NP swabs. The workflow was integrated with an information management system. Self-collected saliva was brought to the reception area by COVID-19 Liaison Officers (CLOs). At the reception department, accreditation ID was linked with the sample ID in the information management system by barcode. Radio-frequency identification (RFID, Toppan Forms Inc., Tokyo, Japan) was incorporated into the information management system for specimen identification. The laboratory information and reporting system (Neo Polalis and Futada System, H.U. Group Research Institute G.K., Tokyo, Japan) allowed for real-time monitoring of the workflow processes of each test, covering the entire process from receipt of the specimen to reporting the result to CLOs, in addition to recording the internal laboratory workflow (Fig 1).

Five laboratories coordinated to conduct testing for various Olympic Games venues. Quantitative CLEIA and PCR tests were also performed in four laboratories outside the Olympic Village, located in Kawasaki, Tokyo, Sapporo, and Iwaki.

Ethical review and approval were waived for this study due to the Act on the Prevention of Infectious Diseases and Medical Care for Patients with Infectious Diseases (the Infectious Diseases Control Law) in Japan.

## Rigorous quality assurance measures

As the Tokyo 2020 Games was an international multi-sport event during a pandemic, quality laboratory practices based on both global and regional perspectives were rigorously implemented [6, 7]. Actions were taken based on discussion and consensus during the development of the International Organization for Standardization standard (ISO/TS 5798) for SARS-CoV-2 detection using nucleic acid amplification methods [6]. To maintain the expected performance of SARS-CoV-2 laboratory tests, meticulous quality assurance measures such as personnel training, competency assessments, assay performance evaluations, and internal quality controls based on cycle threshold values, were implemented. The assay performance was monitored and confirmed by a nationwide external quality assessment of SARS-CoV-2 nucleic acid amplification tests in Japan [7].

## Results

### Test volume and turnaround time

During the Games, all participating athletes (up to 14,000) and others staying in the Olympic Village were screened daily for SARS-CoV-2 infection with the two-step strategy of screening self-collected saliva samples using a CLEIA followed by confirmatory testing of saliva and

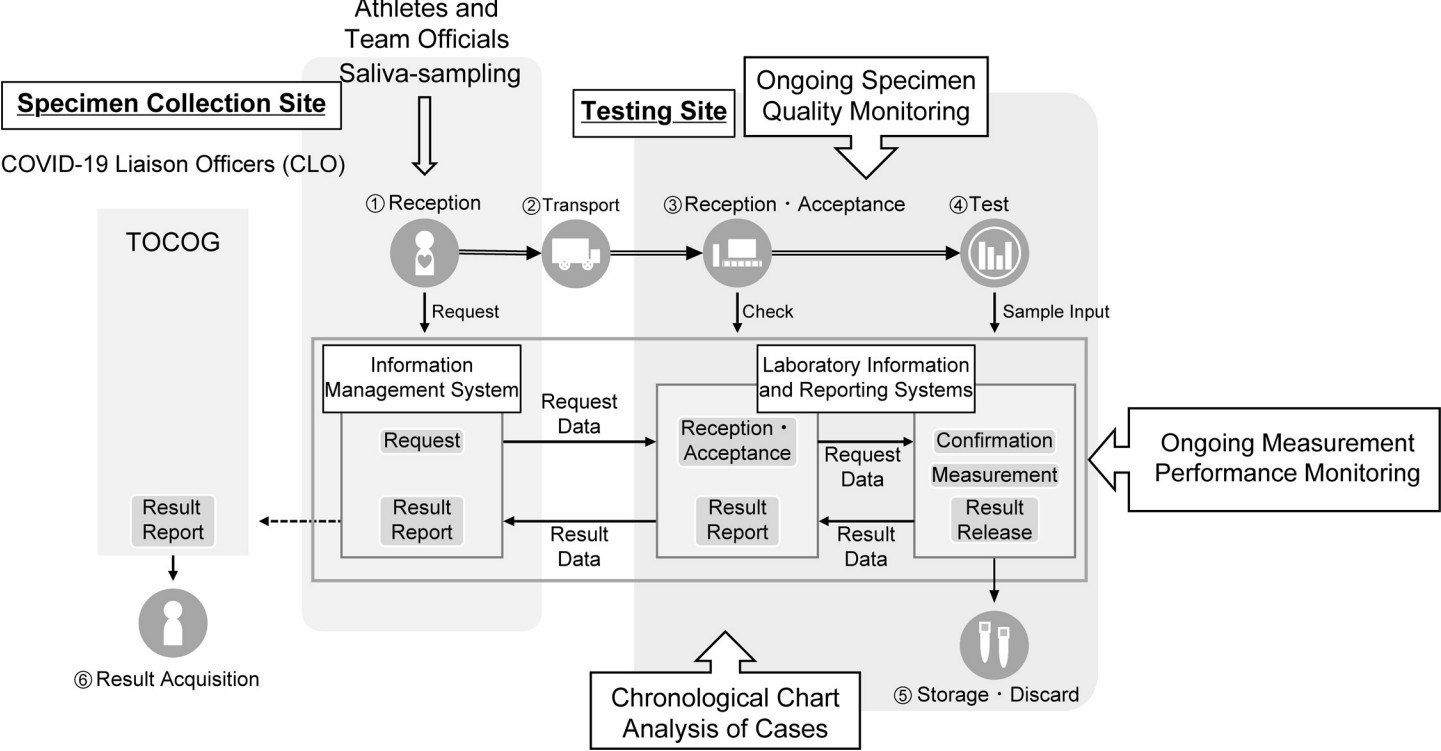

**Fig 1. Workflow quality monitoring.** The workflow was integrated with an information management system. Self-collected saliva was brought to the reception area by COVID-19 Liaison Officers (CLOs). The accreditation ID of personnel in the Olympic Village was linked with radiofrequency identification and sample ID barcodes. This feedback process was used to communicate with the TOCOG about re-training of CLOs and TOCOG staff in appropriate collection techniques, including self-sampling of saliva before eating, specimen refrigeration within 24 hours of sampling, and NP swab collection, enabling real-time corrections of the process.

nasopharyngeal swab samples using PCR. Using a testing system integrated with an information management system covering all steps, from specimen collection to reporting of the results, enabled monitoring of each test and provision of confirmatory test results. A total of 418,506 saliva samples were tested using the quantitative antigen test. Nearly all tests were reported within 12 hours of sample collection. The median TAT was 140 minutes (interquartile range: 103–197 minutes), aligning with the target of a mean TAT of 150 minutes for confirmatory tests.

## Continuous quality improvement

Quality assurance of tests based on both global and regional perspectives was rigorously implemented and continuously monitored. Quality assurance measures included personnel training, competency assessments, assay performance evaluations, and internal quality controls based on cycle threshold values. To ensure the entire examination process from sample collection to the results reporting, we developed and analyzed a chronological chart of cases with positive and inconsistent results, providing a holistic overview of the testing process, confirming test results, and identifying underlying problems including poor specimen quality and false-positive results. Inconsistent results were interpreted and judged whether reasonable as a natural course of infection or suspicious test result due to a possible error in any portion of the testing process.

Particularly, sample collection methods could impact the results; the pre-examination process posed a critical challenge for mass screening. CLOs played a crucial role in instructing

athletes and team officials in proper self-sampling procedures. However, self-sampling in individuals' own rooms in their living quarters had a risk of poor specimen quality due to suboptimal preparation, collection and storage practices. Visual examination of specimens revealed sediments, which caused nonspecific reactions, increasing a risk of false-positive results, rates of re-measurement, and thus prolonging the TAT. We requested the TOCOG to re-educate CLOs about appropriate collection (e.g. prior to eating). To minimize the occurrence of nonspecific reactions due to sediments, the dilution buffer was changed from phosphate buffered saline to a proprietary buffer. As a result, the rate of repeated measurement after re-centrifugation significantly decreased from 8.0% (16/200) to 0.5% (1/200).

We evaluated the quality of 100 specimens using amylase measurement with CicaLiquid AMY (Kanto Chemical Co., Inc., Tokyo, Japan) on a fully automated analyzer (JCA-BM8040, JEOL Ltd., Tokyo, Japan). A high proportion of specimens showed amylase levels under the reference range, suggesting inappropriate storage of specimens at ambient temperature. In the continuous quality improvement of testing during the Games, constant monitoring was implemented to identify potential bias, which allowed real-time actionable feedback to the TOCOG. We used this feedback process to communicate with TOCOG regarding the re-training of CLOs and TOCOG staff in appropriate collection techniques including self-sampling of saliva before eating, specimen refrigeration within 24 hours of sampling, and NP swab collection. This enabled real-time corrections in the testing process. As a result, the activity and performance of athletes for daily training and participation in the Games were ensured. The chronological chart for the individual test results enabled us to confirm in real-time that there were neither infection clusters identified in athletic teams inside the Olympic Village or Games venues due to false-negative results, nor unnecessary withdrawal of athletes from the Games due to false-positive results.

Overall, 11,417 and 4,403 athletes participated in the Olympic and Paralympic Games, respectively. A total of 418,506 screening tests were performed on athletes and team officials during the Games. The screening detected 53 cases, which were later confirmed. The positivity rate was 0.03% (53/15,820) [2].

## Limitations

The testing for SARS-CoV-2 detection to detect infections early was performed, according to the regulation and the criteria specified by the Ministry of Health, and Labour in Japan. Ideally, the quality and competence of the laboratory and the reliability of its testing service would be objectively audited and endorsed by the third-party accreditation. However, in Japan, laboratory accreditation under ISO 15189, Medical Laboratories–Requirements for quality and competence [8], applies to laboratory tests for medical use but not to those for public use such as for SARS-CoV-2 screening tests of participating athletes and others staying in the Olympic Village. Thus, the laboratory was not subjected to accreditation under ISO 15189. The operation of laboratories was built and conducted in alignment with the criteria specified for the central laboratory which had been accredited under ISO 15189 and had a responsibility for the testing for SARS-CoV-2 detection under a contract with the TOCOG.

## Discussion

To detect SARS-CoV-2 infections as soon as possible and to take measures to prevent further spread, a two-step strategy was adopted to meet the mass screening requirements of the event. The integration of a testing system with an information management system covering all steps from specimen collection to reporting of the results enabled confirmatory test results to be provided within 12 hours, achieving our target of a mean TAT of 150 minutes for

confirmatory testing. This integration enabled all athletes to maintain their daily training prior to their events and to participate at the event venue.

The accuracy of testing results can have an impact on the risk of transmission of disease. False-negative results may lead to further transmission of disease. False-positive results would lead to the withdrawal of the athlete from further participation in the Games. Considering the Tokyo 2020 Olympic and Paralympic Games being an international multi-sport event during a pandemic, quality laboratory practices based on both global and regional perspectives were rigorously implemented. To minimize the risk of inaccurate results, we implemented and maintained rigorous quality assurance of the entire process of daily testing for SARS-CoV-2. Quality assurance measures were implemented and continuously monitored, including personnel training, competency assessments, assay performance evaluations, and internal quality controls based on cycle threshold values. The sample collection process was a critical challenge of mass screening. CLOs instructed athletes and team officials on how to perform self-sampling. However, self-sampling in individual rooms posed a risk of poor specimen quality due to suboptimal preparation, collection and storage. To monitor the quality of specimens, we adopted visual inspection and measurement of amylase in saliva. In the continuous quality improvement of testing during the Games, constant monitoring was implemented to identify potential bias, which allowed real-time actionable feedback to the TOCOG.

This case study shows that international major sporting events can be held during major infectious disease epidemics comparable to the COVID-19 pandemic without causing spikes in the case numbers [9]. These results support the approach, advocated by the World Health Organization, that responding to and managing the COVID-19 pandemic requires using all available options, including public health and social measures, robust test-and-trace systems, and vaccination [1]. During the early phase of the COVID-19 pandemic, most mass-gathering sports events used PCR testing for infection control, which required an extended TAT for receiving results and enforcing quarantine [10–12]. Furthermore, PCR has a potential risk of false-positive results, which could have caused unnecessary withdrawal of athletes from the Games. Our experience suggests that continuous quality improvement implemented in the two-step strategy and infection control measures at the Games were successful. Notably, the chronological chart for the individual test results enabled us to confirm in real-time that there were neither infection clusters identified in athletic teams inside the Olympic Village or Games venues due to false-negative results, nor unnecessary withdrawal of athletes from the Games due to false-positive results.

## Conclusion

In conclusion, continuous quality improvement was implemented in the two-step strategy for mass COVID-19 screening and contributed to the success of international major sporting events during the pandemic. The quality practice, systems, and workflows of testing with continuous quality improvement, described in this report could serve as a model for future mass-gathering sporting events during similar major infectious disease epidemics.

## Acknowledgments

We would like to thank the Tokyo 2020 Organizing Committee. The findings and conclusions in this report are those of the authors and do not necessarily represent the official position of the Tokyo 2020 Organizing Committee. The Tokyo 2020 Organizing Committee does not endorse any commercial products or services.

## Author Contributions

**Conceptualization:** Hayato Miyachi, Satomi Asai, Rika Kuroki, Kazuya Omi, Chiaki Ikenoue, Satoshi Shimada.

**Data curation:** Hayato Miyachi, Rika Kuroki, Kazuya Omi, Satoshi Shimada.

**Formal analysis:** Hayato Miyachi, Rika Kuroki, Kazuya Omi, Satoshi Shimada.

**Investigation:** Hayato Miyachi.

**Methodology:** Hayato Miyachi.

**Project administration:** Hayato Miyachi.

**Resources:** Rika Kuroki, Kazuya Omi, Satoshi Shimada.

**Writing – review & editing:** Hayato Miyachi, Satomi Asai, Chiaki Ikenoue, Satoshi Shimada.

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
