## [Decision Letter · Decision Letter 0]

8 Jul 2024

PONE-D-24-17763Continuous quality improvement with a two-step strategy effective for large-scale SARS-CoV-2 screening at the Tokyo 2020 Olympic and Paralympic GamesPLOS ONE

Dear Dr. Miyachi,

Thank you for submitting your manuscript to PLOS ONE. After careful consideration, we feel that it has merit but does not fully meet PLOS ONE’s publication criteria as it currently stands. Therefore, we invite you to submit a revised version of the manuscript that addresses the points raised during the review process.

We look forward to receiving your revised manuscript.

Kind regards,

Etsuro Ito, Ph.D.

Academic Editor

PLOS ONE

Journal Requirements:

2. Thank you for stating the following in the Competing Interests section: "I have read the journal's policy and the authors of this manuscript have the following competing interests: For non-financial conflicts of interest, H.M., S.A., S.S., and C.I. have published several papers about COVID-19 and could have conflicts of interest among healthcare professionals. For financial conflicts of interest, H.M. was a consultant at SRL Laboratories and an adviser for the laboratory in the Olympic and Paralympic Village in Harumi, Tokyo, Japan. H.M. and S.A. were involved in the nationwide external quality assessment of SARS-CoV-2 nucleic acid tests, a project commissioned by the Ministry of Health, Labour and Welfare of Japan. S.A. contributed to studies on SARS-CoV-2 PCR tests developed by DENSO Corp. and KYORIN Pharm. Co., Ltd. that were financially supported by the Japan Agency for Medical Research and Development. R.K. and K.O. rendered laboratory service for athletes and team officials during the Tokyo 2020 Games as a commission project by the Tokyo Organizing Committee of the Olympic and Paralympic Games. S.S. and C. I. worked as staff members for Tokyo 2020 during the Tokyo 2020 Games. The authors report no other potential conflicts of interest."

3. In the online submission form, you indicated that "The data that support the findings of this study are available from the corresponding author upon reasonable request."

Reviewers' comments:

Reviewer's Responses to Questions

**Comments to the Author**

1. Is the manuscript technically sound, and do the data support the conclusions?

Reviewer #1: Partly

Reviewer #2: No

Reviewer #3: Yes

2. Has the statistical analysis been performed appropriately and rigorously? 

Reviewer #1: Yes

Reviewer #2: N/A

Reviewer #3: Yes

3. Have the authors made all data underlying the findings in their manuscript fully available?

Reviewer #1: No

Reviewer #2: No

Reviewer #3: No

4. Is the manuscript presented in an intelligible fashion and written in standard English?

Reviewer #1: No

Reviewer #2: Yes

Reviewer #3: Yes

5. Review Comments to the Author

Reviewer #1: This manuscript is interesting in the field of laboratory medicine. However, the purpose is obscure and completely retrospective fashion. I can not find the novel point in this manuscript. In addition, the conclusion is not new for the researcher engaged in the field.

Reviewer #2: In this study, Miyachi et al. described a process of two-step strategy for large-scale screening of SARS-CoV-2 at the Tokyo 2020 Olympic. Although this paper is submitted as a research article, almost no data presented. Important studies that are the scientific backbone of this airport quarantine system using saliva samples are missing. They should be introduced with comments and references (Yokota, Clin Infect Dis 2021: Yokota, Lancet Microbes 2021).

Reviewer #3: Miyachi et al. describe in this manuscript their experience with large-scale SARS-CoV-2 screening of participants at the Tokyo Olympics, and state that it provides a model for future infectious disease responses to large-scale sporting events. The two-stage strategy proposed and implemented by the authors, using antigen quantification testing prior to PCR, was presumably beneficial, but more information in support of this would be helpful.

Major comments.

(1) Page 7, lines 148-150: Although the mean of TAT is given, I feel that this alone makes it difficult to assess the bias of the values. Would it be possible to show the distribution in the raw data or information on quartiles?

(2) Page 8, lines 179-182: It seems to me that in order to assess the actual extent of disadvantage due to false-positive results, information such as the results of the second PCR test among those who were positive in the first stage of antigen quantification testing is needed. Although the final percentage of positives is shown in page 10, lines 230-231, I think that these should be included in the Result together with the above breakdown.

(3) Page 10, lines 226-227: It is mentioned that the PCR test has a potential risk of false positives, but I think that the risk of false positives for antigen quantification tests should also be mentioned. Rather, to reduce the risk of final false-positives, is the design of antigen quantification as a prior step to the PCR method more important?

Minor comments

(1) Page 3, Line 51: "Tokyo 2020 Games" should be changed to "Tokyo 2020 Olympic and Paralympic Games".

(2) Page 3, Line 63: "instead of PCR screening" alone does not make sense, so the wording should make it clear that it is a comparison with PCR alone.

(3) Page 7, lines 167-171: Can you give the information regarding sample number of quality assessment by amylase measurement?

6. PLOS authors have the option to publish the peer review history of their article (what does this mean?). If published, this will include your full peer review and any attached files.

Reviewer #1: No

Reviewer #2: No

Reviewer #3: No

---

## [Author Response · Author response to Decision Letter 0]

16 Aug 2024

Response to reviewers uploaded in the Attach Files step.

---

## [Decision Letter · Decision Letter 1]

12 Sep 2024

Continuous quality improvement with a two-step strategy effective for mass SARS-CoV-2 screening at the Tokyo 2020 Olympic and Paralympic Games

PONE-D-24-17763R1

Dear Dr. Miyachi,

We’re pleased to inform you that your manuscript has been judged scientifically suitable for publication and will be formally accepted for publication once it meets all outstanding technical requirements.

Kind regards,

Etsuro Ito, Ph.D.

Academic Editor

PLOS ONE

Reviewers' comments:

Reviewer's Responses to Questions

**Comments to the Author**

1. If the authors have adequately addressed your comments raised in a previous round of review and you feel that this manuscript is now acceptable for publication, you may indicate that here to bypass the “Comments to the Author” section, enter your conflict of interest statement in the “Confidential to Editor” section, and submit your "Accept" recommendation.

Reviewer #2: All comments have been addressed

2. Is the manuscript technically sound, and do the data support the conclusions?

Reviewer #2: Partly

3. Has the statistical analysis been performed appropriately and rigorously? 

Reviewer #2: N/A

4. Have the authors made all data underlying the findings in their manuscript fully available?

Reviewer #2: No

5. Is the manuscript presented in an intelligible fashion and written in standard English?

Reviewer #2: Yes

6. Review Comments to the Author

Reviewer #2: The authors responded all the issues raised and the manuscript has been improved. I have no more comments on this

7. PLOS authors have the option to publish the peer review history of their article (what does this mean?). If published, this will include your full peer review and any attached files.

Reviewer #2: No

---

## [Editor Report · Acceptance letter]

16 Sep 2024

PONE-D-24-17763R1 

PLOS ONE

Dear Dr. Miyachi, 

I'm pleased to inform you that your manuscript has been deemed suitable for publication in PLOS ONE. Congratulations! Your manuscript is now being handed over to our production team.

Kind regards, 

on behalf of

Prof. Etsuro Ito 

Academic Editor

PLOS ONE